# Keep It Brief and Targeted: Driving Performance Feedback Report Features to Use with Novice Drivers

Chelsea M. Ward McIntosh [1,*], Elizabeth A. Walshe [1], Shukai Cheng [1], Flaura K. Winston [1,2] and Ellen Peters [3]

1 Center for Injury Research and Prevention, Children's Hospital of Philadelphia, Philadelphia, PA 19104, USA
2 Perelman School of Medicine, University of Pennsylvania, Philadelphia, PA 19014, USA
3 Center for Science Communication Research, University of Oregon, Eugene, OR 97403, USA
* Correspondence: wardc8@chop.edu; Tel.: +1-215-590-1244

**Abstract:** Driving evaluations aim to ensure adequate skills; however, feedback beyond pass/fail is needed for improvement. Therefore, the goal of this study was to inform driving feedback report design to ensure ease of use and understandability while motivating improvements. Participants ages 18–25 years (n = 521) were recruited from CloudResearch Prime Panels to rate one of nine report design conditions with various combinations of five key features: performance summary presence, action plan (AP) length, AP order, AP grading system, and peer comparison presence; they then completed questionnaires. Participants were more motivated to improve when a summary was present ($p = 0.02$); they rated reports easier to use if they had a long AP ($p = 0.01$), a short AP paired with a summary ($p = 0.007$), or an AP with a number grade ($p = 0.016$); and they rated reports easier to understand if they had a short AP ($p = 0.002$) or an AP ordered by worst-to-best performance ($p = 0.05$). These results suggest that feedback reports designed with a performance summary and short, targeted action plan starting with the biggest area for improvement are likely to motivate action to improve driving skills while being easy to use and understand. Future research should evaluate the effect of such a redesigned report on driving outcomes among young drivers.

**Keywords:** feedback; feedback design; young driver; numeracy; information processing; numeric literacy

## 1. Introduction

Driving evaluations are necessary to ensure that drivers have the skills to navigate roadway situations safely (e.g., licensing examinations or fitness-to-drive assessments). However, summative assessments that lead to a dichotomous Pass/Fail—such as driver licensing—do not provide actionable feedback for the driver to know what skills to improve [1]. In order to transform these evaluations into "teachable moments", actionable feedback with personalized strategies for improving skills is needed. This is particularly imperative given the inconsistent findings on the effectiveness of driver education and training: prior research has shown that driver training may improve skills but does not reduce crash risk [2,3], that formal driver's education does reduce crash risk [4,5], but also that mixed results have emerged [6]. Therefore, receiving feedback that includes areas for improvement and a personalized action plan for skill development from an assessment that correlates with crash risk [5] could significantly increase novice driver safety.

While immediate verbal feedback can be beneficial, it could be hard to implement in the driver licensing workflow, and prior work indicates a preference for written feedback reports [7] which allow for re-engagement later [8]. Additionally, as some states have implemented a simulated driving assessment into their driver training and licensing workflow (in addition to the on-road examination for licensure), there is an opportunity to deliver automated personalized feedback reports to all new drivers, informing them of the skills they need to improve to drive safely [9,10].

Prior work has shown that driver performance feedback can reduce driving errors [11], reduce relative risk [12], and improve overall driving performance [13]. This feedback is particularly necessary for newly licensed drivers as crash risk is highest in the first months of licensure then declines over the first year [14], likely as drivers become more experienced. Naturalistic and simulated driving studies demonstrate that young novice drivers make more errors while driving than do older, more experienced drivers [15]; lack of experience is considered the critical/leading causes of crashes [16–18]. Given this critical learning period and the ability to provide driver performance feedback at scale [10], this study sought to examine driver performance feedback report features to optimize report design.

A review of research on feedback report design found most research focusing on writing/grammar (particularly learning English as a second language, e.g., [19]), math performances (e.g., [20]), and medically related activities (such as performing CPR, e.g., [21]), with most driving-specific research being related to autonomous vehicle programming (e.g., [22]). Prior studies have shown that in order to be effective, feedback reports need to be easy to use and motivate the recipient to improve [23]. Feedback should begin with positive information, identify specific areas for improvement, and provide guidance for improvement [24]. However, feedback reports sometimes are presented in ways that are overwhelming or confusing for young drivers; feedback report designers therefore should seek to avoid cognitive burden and highlight the meaning of important information [25,26]. Furthermore, to ensure feedback report recipients at all numeracy levels ("number literacy") can understand the information, presenting fewer options and less information may increase comprehension [25].

Given the lack of research on design features and the importance of this feedback, this study examined driver performance feedback report design by varying report features in theoretically based ways to determine how best to present information in an easy to use, easy to understand manner that also motivates the recipient to improve their driving skills. As prior research shows "less is more" in relaying information [25], two of the feedback report features manipulated in this research were the presence of a performance summary (present/absent) and the length of the individualized action plan (AP; short/long). To see the potential impact of one's numeracy level [27,28], the AP grading system was also manipulated (numeric grade, letter grade, or combination of both). As prior research also suggested putting the positive information first [24], the AP order was manipulated (best-to-worst performance, worst-to-best, or by importance). Lastly, the presence of a peer comparison (present/absent) was also tested as peer comparisons have been shown to increase motivation [26].

In an online survey study sample, these report design features ("conditions") were assessed using three main outcome measures: motivation to improve (how likely they are to think about improving their safe driving behaviors), ease of use (how they like the report and their perceptions of it), and ease of understanding (how accurately they perceive the information). We further tested whether numeracy might moderate any effects of feedback report conditions. Given prior findings, it was hypothesized:

1. Numeracy: More numerate participants (those better able to understand numbers and probability [29]) will find the reports easier to use and easier to understand; they will also have higher motivation to improve than less numerate participants.
2. Summary presence: Reports with a summary present will be easier to understand and lead to higher motivation to improve than a summary absent.
3. AP length: Reports with a short AP will be easier to use, easier to understand, and lead to higher motivation to improve than a long AP.
4. Report length interactions: Reports with a summary absent and a short AP will be the easiest to use, easiest to understand, and lead to highest motivation to improve than any other combination.
5. AP order: Reports with AP ordered best-to-worst will be easier to use, easier to understand, and lead to higher motivation to improve than by worst-to-best or by importance.

6. AP grading: Reports with the letter-number combination grading will be easier to use, easier to understand, and lead to higher motivation to improve than with numbers only or letters only.
7. Peer comparison presence: Reports with peer comparison present will be easier to use, easier to understand, and lead to higher motivation to improve than with peer comparison absent.

## 2. Materials and Methods

### 2.1. Sample

Participants ages 18–25 years old were recruited from CloudResearch Prime Panels [30], which uses Amazon Mechanical Turk (MTurk), in May–June 2021 to rate nine designs of a driver feedback reports and answer questions about the report design via a REDCap [31] survey. Inclusion criteria included: having an IP address in the US, completion of at least 100 prior MTurk human intelligence tasks; and having >90% approval rating. The Children's Hospital of Philadelphia Institutional Review Board determined that this study was exempt per 45 CRF 46.104(d) 2(ii); however, all participants (n = 554) were provided information about the study and asked to agree prior to their participation. Participants were pseudo-randomized using a serial sequential presentation of report layout variations (i.e., the first person saw Condition 1, . . . the ninth saw Condition 9, the tenth saw Condition 1, . . . ) without any constraints to age, gender, or other demographics. Those who agreed (n = 544) completed demographic questions (age in years, gender, license/endorsements status, urbanicity, highest education, race/ethnicity), numeracy questions (subjective and objective), reviewed a feedback report condition, and answered questions about the feedback report. After completing the questions (median ~12.5 min to complete; n = 522), participants were provided $2 compensation. One participant failed more than one attention check question and was excluded from the sample for a final sample of n = 521 (57% female) for the main analyses. The sample for the numeracy analyses (n = 496) was smaller as the objective numeracy questions (appearing after the feedback report condition questions) were entirely skipped by some participants (n = 26) who completed all other measures.

### 2.2. Driver Performance Feedback Report Designs

Participants were pseudo-randomly assigned to one of nine feedback report conditions that were developed using variations of key design features in length and grading. All information, when present, was the same in each condition (e.g., the same grade on a specific item, identical wording); in other words, these reports were not individualized or personalized reports as the goal of the study was to evaluate perceptions of report designs. After reviewing the report design condition (which remained on screen), participants were asked the same questions.

2.2.1. Report Design Features

The driver performance feedback reports contained three main sections of interest:

1. Driving performance summary: This bulleted list provided an overall summary of driving performance, such as "You managed your speed well" and "You followed vehicles too closely, increasing your chances of rear-ending them". The presence or absence of the summary was manipulated across conditions; participants with summary present received identical information.
2. Action plan (AP) with results: This table of skills feedback included domains of safe driving, their definitions, and a grade with suggestions for improvement and links to training materials. The number of domains shown, order of appearance, and grading system varied. In the long AP, eight domains were shown (speed management, road positioning, gap selection, managing blind spot, hazard anticipation and response, attention maintenance, communication/right of way, and vehicle control), while the short AP had four domains (speed management, gap selection, managing blind spot,

and attention maintenance). For the order of appearance, this was by grade best-to-worst, by grade worst-to-best, or by order of domain importance (i.e., gap selection always first). Grading was either by letter grade only, numerical grade only, or a combination of number and letter grade and were the same or equivalent for each domain (e.g., speed management was always A, 96, or 96-A).

3.  Peer comparison: This sentence appeared next to the overall grade and improvement opportunity and stated, "Of all the drivers who completed the virtual driving test in your peer group, 65% drove the virtual driving test safer than you". The presence or absence of the peer comparison was manipulated across conditions; participants with peer comparison present received identical information.

### 2.2.2. Experimental Conditions

The inclusion of the summary and peer comparison sections and the AP format (length, order, grading) were manipulated in order to compare report design features and combinations of such, resulting in nine layout conditions (see Table 1). For example, Condition 1 had a summary (present) and peer comparison (present) with an 8-item AP (long) ordered best-to-worst (best-worst) by letter grade (letter).

**Table 1.** Feedback report conditions.

| Condition | Summary | AP Length | AP Order | AP Grading | Peer Comparison |
|-----------|---------|-----------|----------|------------|-----------------|
| 1 | Present | Long | Best-Worst | Letter | Present |
| 2 | Present | Short | Best-Worst | Letter | Present |
| 3 | Absent | Long | Best-Worst | Letter | Present |
| 4 | Absent | Short | Best-Worst | Letter | Present |
| 5 | Present | Long | Worst-Best | Letter | Present |
| 6 | Present | Long | Importance | Letter | Present |
| 7 | Present | Long | Best-Worst | Letter | Absent |
| 8 | Present | Long | Best-Worst | Number | Absent |
| 9 | Present | Long | Best-Worst | Combination | Absent |

AP = Action Plan.

### 2.3. Questionnaires

There were three main question topics: demographics, feedback report, and numeracy. All questions were in the same order for all participants and all questions required a response (i.e., participants were not able to skip questions).

### 2.3.1. Demographics Questions

Basic demographics questions were asked of each participant, including age in years, gender, race/ethnicity (e.g., Black/African American, Hispanic [32]), type of area residing in (e.g., urban, rural), highest education completed (e.g., less than high school degree, more than a 4-year college degree), and driver's license (e.g., permit, unrestricted) and endorsement statuses (e.g., motorcycle, commercial). One yes/no question also asked about using a calculator or looking up answers for the numeracy questions.

### 2.3.2. Feedback Report Questions

Questions were developed for this study to specifically target three main areas of interest related to the feedback report designs. The questions were also worded to reflect these reports not being personalized (e.g., "If this were about my driving, I would be motivated to improve or practice my driving").

1.  Easy to use: These questions checked the information was clearly displayed and the report was easy enough to use they would recommend for others (e.g., "The information in this report was easy to understand"). The questions were on a 7-point Likert-type scale (i.e., strongly disagree to strongly agree) with median responses used.

2. Easy to understand: These questions checked participants could navigate the form and find critical information (e.g., "Which of these driving skills does the report indicate the driver needs the most improvement on?"). The questions were multiple-choice questions transformed into correct/incorrect answers.

3. Motivation to improve: These questions checked the report made the participants reflect on their driving skills (e.g., "This report makes me want to be a safer driver"). The questions were on a 7-point Likert-type scale (i.e., strongly disagree to strongly agree) with median responses used.

Four questions for each of the main areas of interest were asked. In addition, three attention-check questions (e.g., "If you read this, select Strongly Disagree") were interspersed throughout the feedback report questions to determine participant attention, with two on a 7-point Likert-type scale and one multiple choice, which were all transformed into correct/incorrect. Any participant with more than one attention check question incorrect was not included in analysis (n = 1).

### 2.3.3. Numeracy Scale

Participants were asked both subjective and objective numeracy questions. The subjective numeracy scale was presented after the demographics questions and prior to viewing the feedback report; while it was measured, it is outside the scope of the present paper. The objective numeracy scale, adapted from prior numeracy measures [33,34], appeared after the feedback report questions, and included nine fill-in-the-blank word problems of objective numeracy (e.g., "If the chance of getting a disease is 60 out of 300, this would be the same as having what percent chance of getting the disease?"). The questions were presented in four order variations across the nine conditions, resulting in a total of 36 combinations of condition/numeracy question order. Responses were transformed into correct/incorrect, and the total sum was used (score range 0–9).

### 2.4. Analysis

Analyses were conducting using SAS. Basic descriptive statistics of the sample and outcome measures were first reviewed. The Likert-type scale responses were transformed using the median of median responses and centered the median (easy to use and motivation to improve) or mean (easy to understand); this allowed for a series of 1-way ANOVAs to assess significant differences in each of the dependent variables (easy to use, easy to understand, and motivation to improve) across conditions (e.g., AP length). A follow-up ANCOVA was conducted for testing interactions between independent variables, specifically summary presence, AP length, and objective numeracy score.

## 3. Results

### 3.1. Sample Descriptives

The demographic profiles of participants were similar across all conditions, including age, gender, license status, urbanicity, highest education completed, ethnicity, and race (see Table 2). There were no differences in ratings that could be attributed to demographic characteristics except for gender; females overall had higher motivation to improve ($p < 0.0065$) and found the reports easier to use ($p < 0.002$) than males.

Numeracy was also significantly associated with sex, with males outperforming (having more numerate than) females (males average: 4.77; females average: 4.24; $p = 0.0033$). We also tested if a participants' numeracy had an impact on their motivation to improve or how easy to use and easy to understand they perceived the report. To do so, responses from all conditions were combined. The more numerate participants found the reports easier to understand ($p = 0.001$) and use ($p = 0.02$); numeracy did not predict motivation to improve, however. The sample was not large enough to test further covariations related to feedback report design.

**Table 2.** Sample demographics.

| Demographic | n | % |
|---|---|---|
| Gender | | |
| Female | 293 | 56.24 |
| Male | 211 | 40.50 |
| Non-binary or Other | 17 | 3.26 |
| | | |
| Age | | |
| 18 | 11 | 2.11 |
| 19 | 33 | 6.33 |
| 20 | 58 | 11.13 |
| 21 | 72 | 13.82 |
| 22 | 102 | 19.58 |
| 23 | 94 | 18.04 |
| 24 | 101 | 19.39 |
| 25 | 50 | 9.60 |
| | | |
| License Status | | |
| No permit/license | 20 | 3.84 |
| Permit | 32 | 6.14 |
| Restricted/junior license | 40 | 7.68 |
| Unrestricted license | 428 | 82.15 |
| License endorsement [1] | 23 | 4.41 |
| | | |
| Urbanicity | | |
| Urban | 187 | 35.89 |
| Suburban | 277 | 53.17 |
| Rural | 57 | 10.94 |
| | | |
| Highest Education Completed | | |
| Less than high school | 4 | 0.77 |
| High school degree | 85 | 16.31 |
| Some college/trade school | 199 | 38.20 |
| 4-year degree | 198 | 38.00 |
| More than 4-year degree | 34 | 6.53 |
| | | |
| Ethnicity | | |
| Asian | 73 | 14.01 |
| Black/African American | 55 | 10.56 |
| Hispanic | 32 | 6.14 |
| Native American/Alaskan Native | 0 | 0.00 |
| Native Hawaiian/Other Pacific Islander | 2 | 0.38 |
| White/Caucasian | 313 | 60.08 |
| Other | 1 | 0.19 |
| More than one race | 45 | 8.64 |

[1] Commercial endorsement and/or motorcycle endorsement.

*3.2. Report Descriptives*

Across the conditions, on average participants were motivated to improve (min median of all conditions: 5.26; max median: 6.02; range 1–7). They also reported that the report was easy to use (min median: 5.26, max median: 6.25; range 1–7) and easy to understand (min average: 0.855, max average: 0.944; range 0–1). See Table 3.

**Table 3.** Condition performance.

| Condition | n | Motivation to Improve | Easy to Use | Easy to Understand |
|---|---|---|---|---|
| | | mean of medians | mean of medians | mean accuracy |
| 1 | 57 | 5.66 | 5.79 | 0.855 |
| 2 | 59 | 5.88 | 6.03 | 0.915 |
| 3 | 62 | 5.61 | 5.79 | 0.863 |
| 4 | 59 | 5.26 | 5.26 | 0.944 |
| 5 | 58 | 5.89 | 5.82 | 0.933 |
| 6 | 58 | 5.98 | 6.12 | 0.888 |
| 7 | 57 | 5.82 | 5.94 | 0.873 |
| 8 | 54 | 6.02 | 6.25 | 0.863 |
| 9 | 57 | 5.73 | 5.76 | 0.907 |

### 3.3. Report Length Analyses

To understand the impact of the overall length of the report, summary presence/absence, AP length (long, short), and the interaction between these factors were tested using an ANCOVA (see Table 4). A main effect existed of summary presence: participants who viewed a report with a summary present (vs. absent) were more motivated to improve ($p = 0.02$); ease of use and ease of understanding did not differ. There was also a main effect of AP length: participants who saw a short AP (vs. long AP) perceived the report as easier to understand ($p = 0.002$) but less easy to use ($p = 0.01$); motivation to improve did not differ. Summary presence/absence and AP length had a significant interaction: participants who viewed a report with a short AP and summary present (vs. absent) found it easier to use ($p = 0.007$; present = 0.21, absent = 0.57), but with the long AP, no differences between summary present or absent; there were also no other differences.

**Table 4.** Report length analyses (ANCOVA).

| Feature | Conditions | n | Motivation to Improve Mean [1] | p | Easy to Use Mean [1] | p | Easy to Understand Mean [1] | p |
|---|---|---|---|---|---|---|---|---|
| Report Length [2] | | | | | | | | |
| Summary Present | 1 + 2 | | 0.02 | 0.02 * | 0.06 | 0.90 | −0.01 | 0.45 |
| Summary Absent | 3 + 4 | | −0.32 | | −0.33 | | 0.01 | |
| AP Short | 2 + 4 | 237 | −0.19 | 0.65 | −0.21 | 0.01 * | 0.04 | 0.002 * |
| AP Long | 1 + 3 | | −0.12 | | −0.06 | | −0.03 | |
| Summary * AP | | | | NA | | 0.007 * | | NA |
| | | | | | | | | |
| AP Order [3] | | | | | | | | |
| Best-to-Worst | 1 | | −0.10 | | −0.07 | | −0.04 | |
| Worst-to-Best | 5 | 173 | 0.14 | 0.19 | −0.03 | 0.21 | 0.04 | 0.05 * |
| Importance | 6 | | 0.23 | | 0.27 | | −0.01 | |
| | | | | | | | | |
| AP Grading [3] | | | | | | | | |
| Letter | 7 | | 0.07 | | 0.09 | | −0.02 | |
| Number | 8 | 168 | 0.27 | 0.27 | 0.40 ** | 0.06 | −0.03 | 0.45 |
| Combination | 9 | | −0.02 | | −0.08 ** | | 0.01 | |

[1] Centered means used. [2] Two-way ANCOVA used. [3] One-way ANOVA used. * Significant result. ** Ad hoc *t*-test significant ($p = 0.016$) with Bonferroni correction.

### 3.4. AP Order

A one-way ANOVA was used to test the effect of AP order (by grade best-to-worst, by grade worst-to-best, or by importance); see Table 4. There was a main effect of AP order: participants who viewed a report with an AP ordered by performance worst-to-best had the highest score on ease of understanding, and those who saw a report ordered best-to-worst had the lowest ease of understanding ($p = 0.05$); ease of use and motivation to improve did not differ. Ad hoc *t*-tests were performed to test differences between each pair of AP order with Bonferroni corrections and found no differences between conditions.

*3.5. AP Grading*

To test the effect of AP grading system (letter grade, number grade, or combination), a one-way ANOVA was used and did not find any main effect on any of the three outcomes. Ad hoc *t*-tests were performed to test differences between each pair of AP grading with Bonferroni corrections and showed significant differences between number and combination grading systems in ease of use ($p = 0.016$); that is, participants who only had the number grade reported the form was easier to use than participants who had both a number and letter grade; there were no other differences.

*3.6. Peer Comparison*

Peer comparison presence/absence was also tested using a one-way ANOVA with Conditions 1 and 7; participants had no differences in their motivation to improve ($p = 0.34$), ease of use ($p = 0.42$), or ease of understanding ($p = 0.48$).

*3.7. Summary*

In summary, this study tested which feedback report design features are easy to use, easy to understand, and motivate to improve. Participants in the study were motivated to improve when a performance summary was present, but no other conditions impacted motivation. Participants found it easiest to use reports with a long AP, with a short AP paired with summary presence, and with an AP graded by number only. Participants also reported easier understanding of reports with a short AP and an AP ordered by performance from worst-to-best.

**4. Discussion**

This study was designed to inform driver feedback report design with the goal of motivating recipients to improve their performance. By manipulating the features of the report, we found that while some of our hypotheses were confirmed, others went unsupported or were contradicted. As expected, we found that having a summary present led to more motivation to improve and that having a short, focused action plan increased understanding of the report. However, the shorter action plan by itself was more difficult to use, while pairing it with a summary made it the easiest to use. This finding seems to confirm those from other areas of research that keeping information brief improves comprehension [25]. The study findings recommend a design that includes a performance summary and short action plan for an easy-to-use and easy-to-understand feedback report that motivates for improvement.

This study also reviewed other design aspects of the feedback report beyond length. Although we hypothesized that having the action plan presented from best performance to worst performance would be the easiest to use and understand, and would motivate the most for improvement, we found that presenting items from worst-to-best was the easiest to understand. Thus, it is recommended to use a design that presents the main areas for improvement first, which is contrary to education research indicating that feedback should always begin with the positive [24].

We had also expected grades reported in a letter-number combination would perform best overall but found that numbers-only was easiest to use. Interestingly, participants who were more numerate (i.e., understand numbers better) also found the reports easiest to use. While we expected a peer comparison presence to have an impact, no significant differences emerged based on whether a peer comparison was present.

The findings presented here are limited in their effect by the hypothetical nature of the study and the small sample size per condition (~n = 58 per condition). These feedback reports are intended for novice drivers, in particular teens and adolescents 16–19 years old; due to the limitations of online research, this target age range was underrepresented. As novice drivers can be any age, further testing with a wider age range would identify whether the report is more generalizable. Further research, including in a real driving setting and with larger samples, also is needed to confirm these findings.

Additionally, these feedback reports were not personalized to the participants—that is, they did not receive feedback based on their own driving performance. It is possible the results would be different if the feedback was personally meaningful to participants. In the future, testing these feature conditions with personalized reports for teen/adolescent drivers may result in more meaningful results. Using personalized reports would also allow for testing the actual impact of the reports and improvements in the drivers' performance. This step is particularly important as prior research shows that disagreeing with feedback can lead to dismissing it [35]. Further design testing could also include both goal setting [36] and video clips [37] along with the automated driving performance feedback which may improve performance even further.

Lastly, the questionnaires used to measure ease of use, ease of understanding, and motivation to improve were developed for this study. The results of this research could further be strengthened by incorporating validated measures, such as the System Usability Scale [38], or adapting existing scales, such as the Health Literacy Measure for Adolescents [39].

While the future of driver training and education may change as autonomous vehicles become more ubiquitous, need will exist for this or similar training and assessment [40] for years to come. When delivered to novice drivers before or at the time of licensure, written performance feedback reports could lead to improved safe driving skills. More research is warranted to distinguish if the design features identified here remain most important when used with the target age range for new drivers and using personally relevant information. Still, these features should be considered, along with numeracy, when designing feedback reports so they are more usable, understandable actionable feedback reports that motivate adolescents to implement more effective strategies for safer driving and potentially reduce fatal crash risk. While focused on driving performance, these feedback report design features should be incorporated and studied in other areas as well.

**Author Contributions:** Conceptualization, C.M.W.M., F.K.W. and E.P.; methodology, C.M.W.M. and E.P.; software, C.M.W.M.; validation, C.M.W.M., E.P. and E.A.W.; formal analysis, S.C., E.A.W. and C.M.W.M.; investigation, C.M.W.M. and E.P.; resources, C.M.W.M. and F.K.W.; data curation, C.M.W.M.; writing—original draft preparation, C.M.W.M.; writing—review and editing, E.A.W., S.C., F.K.W. and E.P.; visualization, S.C. and E.A.W.; supervision, E.P., E.A.W. and F.K.W.; project administration, C.M.W.M.; funding acquisition, F.K.W. All authors have read and agreed to the published version of the manuscript.

**Funding:** This research was funded by a gift from New Jersey Manufacturers (NJM) Insurance Group to CHOP.

**Institutional Review Board Statement:** Ethical review and approval were waived for this study due to meeting the criteria for exemption per 45 CFR 46.104(d) 2(ii).

**Informed Consent Statement:** Informed consent was obtained from all subjects involved in the study.

**Data Availability Statement:** Data is available upon request.

**Acknowledgments:** The authors would like to acknowledge the contributions made by Research Assistants Sarah O'Brien and Emily Brown.

**Conflicts of Interest:** Flaura Winston and CHOP own part of Diagnostic Driving, Inc., the company whose driving simulation feedback report template was utilized for this study. The results of this study were reviewed by Ellen Peters from University of Oregon who does not have reporting obligations to CHOP, Diagnostic Driving, or Winston. Diagnostic Driving was not involved in any aspect of the study other than providing a report template.

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
