# Peer review of "Keep It Brief and Targeted: Driving Performance Feedback Report Features to Use with Novice Drivers"

_adolescents, doi:10.3390/adolescents2040035_

Round 1

Reviewer 1 Report

The article addresses an area related to road safety including the formation of future traffic users. The research group is very narrow and it would be useful to do a study with a wider age range. In addition, it is also worth asking drivers about the future of transport, i.e. autonomous vehicles. Soon the scope of driving courses will certainly be changed due to technological developments.

Can other survey tools be used for your research? And if so, which ones?

Reviewer 2 Report

A nice paper looking at a detail of driver feedback as part of the licensing system. The only part I feel that does need to be expanded is lines 325-326 "When delivered to novice drivers before or at the time of licensure, written performance feedback reports could lead to improved safe driving skills." This is strictly correct as the authors have used 'could', but I would suggest that this claim is tempered further with a note about the very poor record of driver education and training approaches as safety improvements. A good place to start would be the Cochrane review on school-based education and subsequent citing literature: School‐based driver education for the prevention of traffic crashes - Roberts, IG - 2001 | Cochrane Library

The point here is that although this work is about licensing, driver education is usually posited as using similar mechanisms to prompt safer behaviour. So in short, I think acknowledgement of the driver education literature in terms of its poor record on actual safety outcomes (as well as the relevant work from people like Simons-Morton and Mirman) in this work is warranted. 
